# Diet modifies the association between alcohol consumption and severe alcohol-related liver disease incidence

Fanny Petermann-Rocha [1,2,9], Ziyi Zhou[3], John C. Mathers [4], Carlos Celis-Morales [1,5,6], David Raubenheimer[7], Naveed Sattar [1], Jill P. Pell [3], Ewan Forrest[8,9] & Frederick K. Ho [3,9] ✉

It is elusive why some heavy drinkers progress to severe alcohol-related liver disease (ALD) while others do not. This study aimed to investigate if the association between alcohol consumption and severe ALD is modified by diet. This prospective study included 303,269 UK Biobank participants. Alcohol consumption and diet were self-reported. The diet score was created from 4 items selected using LASSO. Cox proportional hazard model showed that the diet score was monotonically associated with severe ALD risk, adjusted for sociodemographics, lifestyle factors, and alcohol consumption. Relative excess risk due to interaction analysis indicated that having a higher ALD diet score and a higher alcohol consumption simultaneously confers to 2.44 times (95% CI: 1.06-3.83) higher risk than the sum of excess risk of each factor. In this work, we show that people who have a poor diet might be more susceptible to severe ALD due to alcohol consumption.

Alcohol-related liver disease (ALD) results from a liver injury associated with alcohol consumption and is frequently linked to psychiatric comorbidities[1]. Along with alcohol use disorders, ALD accounts for over a quarter (26.3%) of all alcohol-attributable mortality worldwide[2]. In the UK, ALD was the cause of nearly 78% of alcohol-specific deaths in 2021[3]. The disease encompasses a spectrum of conditions, starting from the milder and reversible alcoholic hepatic steatosis (fatty liver). It can, at times, progress to alcoholic hepatitis, alcoholic fibrosis, and culminate in the advanced and irreversible stage of alcoholic-related cirrhosis[4,5].

Intuitively, alcohol consumption is the necessary factor for developing ALD, and the leading risk factor for the onset of severe ALD, e.g., those that require hospitalisation or those that lead to death. Previous studies have indicated that ALD incidence and mortality are proportional to the amount of alcohol consumed[6–9]. However, only 10–20% of individuals with chronic heavy alcohol use develop cirrhosis or alcoholic hepatitis[10]. The association between alcohol consumption and ALD can be influenced by metabolic, genetic, immunological, and environmental factors[4,5].

Among these potential moderators, diet is one of the more modifiable. The impact of diet on liver diseases has been extensively studied for non-alcoholic liver diseases (NAFLD)[11–18]. However, few studies have delved into the influence of diet (either in terms of individual or combined foods and nutrients) on the onset of severe ALD[19,20]. Regarding individual nutrients, magnesium deficiency appears to elevate the risk of alcohol-related liver injury[19], while low vitamin D concentrations have been linked to liver damage and mortality in ALD[20]. Coffee consumption might

[1]School of Cardiovascular and Metabolic Health. University of Glasgow, Glasgow, UK. [2]Centro de Investigación Biomédica, Facultad de Medicina, Universidad Diego Portales, Santiago, Chile. [3]School of Health and Wellbeing, University of Glasgow, Glasgow, UK. [4]Human Nutrition & Exercise Research Centre, Centre for Healthier Lives, Population Health Sciences Institute, Newcastle University, Newcastle upon Tyne, UK. [5]Human Performance Laboratory, Education, Physical Activity and Health Research Unit, Universidad Católica del Maule, Talca, Chile. [6]Centro de Investigación en Medicina de Altura (CEIMA), Universidad Arturo Prat, Iquique, Chile. [7]Charles Perkins Centre and School of Life and Environmental Sciences, The University of Sydney, Sydney, Australia. [8]Department of Gastroenterology, Glasgow Royal Infirmary; University of Glasgow, Glasgow, UK. [9]These authors contributed equally: Fanny Petermann-Rocha, Ewan Forrest, Frederick K. Ho. ✉e-mail: Frederick.ho@glasgow.ac.uk

also offer a protective effect against ALD progression and alcohol-related cirrhosis[21,22].

In this work, we investigated if the association between alcohol consumption and severe ALD, as indicated by hospital admission and death attributed to ALD, is modified by diet using data from the UK Biobank prospective cohort study. We show that having a higher ALD-related dietary score was associated with higher risk of severe ALD after adjustment of alcohol intake and other confounders, and people who have a poor diet might be more susceptible to severe ALD due to alcohol consumption.

## Results

### Overall characteristics of the included population

After excluding participants with liver disease or drug consumption at baseline (Supplementary Table 1), missing data for the diet score created or any covariate, and those who developed the disease event within the first two years of follow-up, 303,269 participants were included in the main analyses (Fig. 1). Overall, the included participants had a mean age of 56.4 years, were more likely to be women, to never smoke, and a higher proportion had hypertension/high blood pressure (Table 1). In terms of the individual diet and alcohol categories created, and compared with the healthiest group (lower ALD dietary risk score and <14 units of alcohol per week), those in the most at-risk group (higher ALD dietary risk score and ≥14 units of alcohol per week) were younger, more likely to be men, to smoke previously and to drink alcohol daily or almost daily. They also had a higher prevalence of hyperglycaemia, high triglycerides, central obesity, and hypertension (Table 1). The marginal difference in total alcohol consumption and alcohol drinking frequency between ALD dietary risk score groups were of small effect sizes, with Cohen's *d* 0.09 and Phi coefficient 0.12.

### Associations between diet score and alcohol consumption categories on severe ALD risk

Over a median follow-up of 10.7 years (interquartile range: 10.0–11.4), 539 (0.2%) participants were diagnosed with severe ALD. Of these, 69 (12.8%) were alcohol-related hepatitis and 266 (49.4%) alcohol-related cirrhosis. Figure 2 shows the nonlinear association between the continuous diet score and ALD incidence, adjusted for age, sex, deprivation, ethnicity, smoking, physical activity, components of the metabolic syndrome, units/week of alcohol consumption, and frequency of drinking. The left panel shows the association in all participants. There was no evidence for nonlinearity on the logarithm scale, and the association appeared to show a monotonic increasing trend. For instance, participants with a diet score of 0.3 had about 1.5 times higher risk of severe ALD, while those with a score of 0.6 had about 2 times higher risk compared with the lowest value of the diet score. The right panel of the figure shows the analysis stratified by alcohol consumption (lower vs. increasing & higher risk). The association between diet score and severe ALD was stronger in the group with higher alcohol consumption. In fact, among people with lower risk of alcohol consumption, the association between diet score and severe ALD was only significant when diet score >0.6.

Table 2 shows the joint associations of ALD diet score and alcohol consumption categories with severe ALD risk. The risk of severe ALD was highest among people who had higher alcohol risk and a higher ALD diet score (HR 14.20; 95% CI 8.24–24.50). There was significant additive interaction (RERI 2.44; 95% CI 1.06–3.83), indicating that having higher alcohol risk and a higher diet score was associated with 2.44 times higher severe ALD risk than the sum of the excess risk of the two factors (Fig. 3). Similar findings were found by sexes, where the interaction was stronger in women (RERI 3.22, 95% CI 0.78–5.65) than men (RERI 2.23; 95% CI 0.26–4.19). The associations and interaction showed similar conclusions when alcohol consumption was modelled as a continuous variable (Supplementary Table 4). Analyses for specific ALD outcomes (cirrhosis and hepatitis) are shown in Supplementary Table 5, which also showed similar patterns of associations and interactions, even though the analysis for hepatitis was underpowered.

Finally, assuming causality after adjusting for the same variables, an above-median diet score was attributed to 28.8% (95% CI: 16.0%–39.6%) of severe ALD in the UK Biobank population, while alcohol consumption was attributed to 67.1% (95% CI: 61.5%–71.9%) (Table 3). Similar findings were found when alcohol consumption and diet score were modelled as continuous variables (Supplementary Table 6).

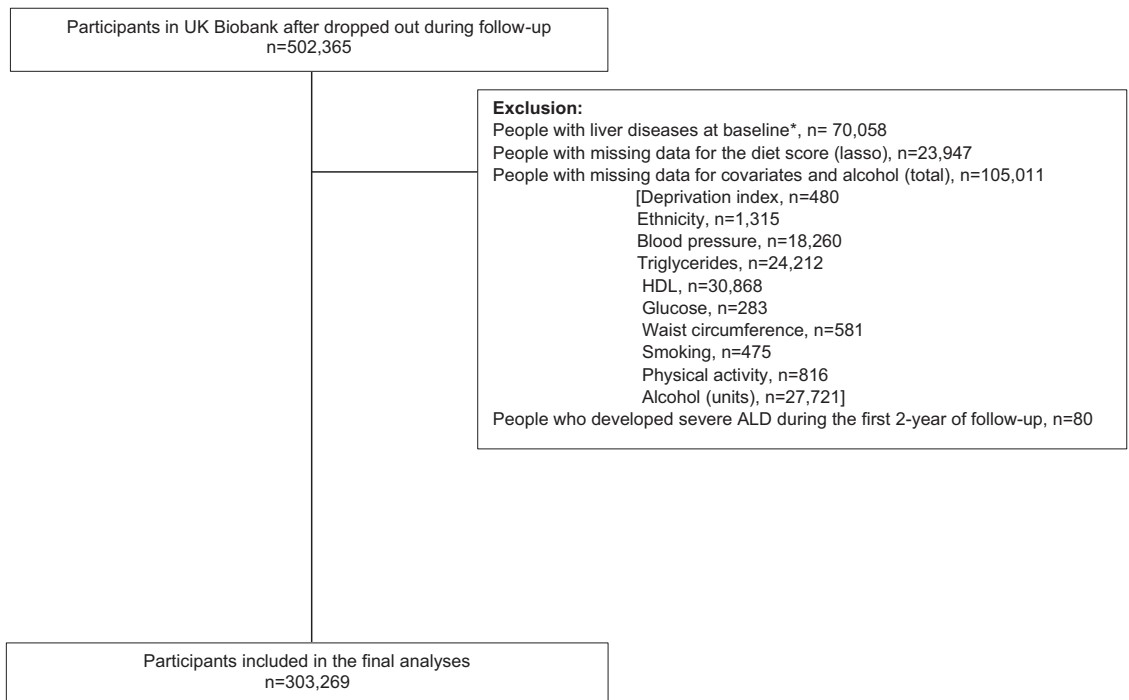

**Fig. 1 | Participant flowchart.** Flowchart showing the selection of participants into this study.

**Table 1 | Baseline characteristics by diet score and alcohol consumption categories**

| | Total | Lower ALD dietary risk score and <14 units of alcohol (lower risk) | Lower ALD dietary risk score and ≥14 units of alcohol (increasing and higher risk) | Higher ALD dietary risk score and <14 units of alcohol (lower risk) | Higher ALD dietary risk score and ≥14 units of alcohol (increasing and higher risk) |
|---|---|---|---|---|---|
| n, (%) | 303,269 (100) | 98,343 (32.4) | 50,874 (16.8) | 78,475 (25.9) | 75,577 (24.9) |
| Baseline age (years), mean (SD) | 56.4 (8.1) | 57.1 (8.0) | 56.9 (7.9) | 55.8 (8.3) | 55.8 (8.0) |
| Sex, n (%) | | | | | |
| Women | 159,568 (52.6) | 66,967 (68.1) | 17,907 (35.2) | 50,734 (64.6) | 23,960 (31.7) |
| Men | 143,701 (47.4) | 31,376 (31.9) | 32,967 (64.8) | 27,741 (35.4) | 51,617 (68.3) |
| Deprivation index, mean (SD) | -1.44 (3.0) | -1.65 (2.9) | -1.86 (2.7) | -1.01 (3.2) | -1.32 (3.02) |
| Ethnicity, n (%) | | | | | |
| White | 287,693 (94.9) | 93,850 (95.4) | 50,224 (98.7) | 69,921 (89.1) | 73,698 (97.5) |
| Others | 15,576 (5.1) | 4493 (4.6) | 650 (1.3) | 8554 (10.9) | 1879 (2.5) |
| Smoking status, n (%) | | | | | |
| Never | 167,890 (55.4) | 66,004 (67.1) | 24,981 (49.1) | 46,170 (58.8) | 30,735 (40.7) |
| Previous | 105,625 (34.8) | 27,630 (28.1) | 22,121 (43.5) | 23,371 (29.8) | 32,503 (43.0) |
| Current | 29,754 (9.8) | 4709 (4.8) | 3772 (7.4) | 8934 (11.4) | 12,339 (16.3) |
| Type of physical activity, n (%) | | | | | |
| Walking for pleasure (not as a means of transport) | 219,034 (72.2) | 73,857 (75.2) | 39,292 (77.2) | 53,038 (67.6) | 52,847 (69.9) |
| Other exercises (e.g.: swimming, cycling, keep fit, bowling) | 37,418 (12.3) | 11,538 (11.7) | 6417 (12.6) | 9619 (12.3) | 9844 (13.0) |
| Strenuous sports | 2451 (0.8) | 596 (0.6) | 506 (1.0) | 525 (0.7) | 824 (1.1) |
| Light DIY (e.g.: pruning, watering the lawn) | 19,575 (6.5) | 5692 (5.8) | 2296 (4.5) | 6229 (7.9) | 5358 (7.1) |
| Heavy DIY (e.g.: weeding, lawn mowing, carpentry, digging) | 7526 (2.5) | 1902 (1.9) | 998 (2.0) | 2145 (2.7) | 2481 (3.3) |
| None of the above or prefer not to answer | 17,265 (5.7) | 4758 (4.8) | 1365 (2.7) | 6919 (8.8) | 4223 (5.6) |
| Alcohol consumption, units/week, mean (SD) | 16.4 (18.6) | 5.2 (4.3) | 27.8 (15.0) | 5.2 (4.4) | 34.9 (21.6) |
| Alcohol frequency intake, n (%) | | | | | |
| Daily or almost daily | 67,386 (22.2) | 5523 (5.6) | 21,930 (43.1) | 4173 (5.3) | 35,760 (47.3) |
| 3–4 times a week | 77,698 (25.6) | 17,101 (17.3) | 20,636 (40.6) | 12,630 (16.1) | 27,331 (36.2) |
| Once or twice a week | 84,861 (28.0) | 36,314 (36.9) | 8297 (16.3) | 27,792 (35.4) | 12,458 (16.5) |
| 1–3 times a month | 12,875 (4.3) | 7059 (7.1) | 11 (0.0) | 5777 (7.4) | 28 (0) |
| Special occasions only | 36,118 (11.9) | 19,656 (20.2) | 0 | 16,462 (21.0) | 0 |
| Never | 24,331 (8.0) | 12,690 (12.9) | 0 | 11,641 (14.8) | 0 |
| Hyperglycaemia/diabetes (yes), n (%) | 44,575 (14.7) | 13,918 (14.2) | 6994 (13.8) | 11,866 (15.1) | 11,797 (15.6) |
| Low HDL (yes), n (%) | 56,551 (18.6) | 22,164 (22.5) | 5742 (11.3) | 19,919 (25.4) | 8726 (11.6) |
| High triglycerides (yes), n (%) | 118,943 (39.2) | 35,074 (35.7) | 19,807 (38.9) | 30,784 (39.2) | 33,278 (44.0) |
| Central obesity (yes), n (%) | 96,293 (31.8) | 28,469 (29.0) | 12,990 (25.5) | 29,190 (37.2) | 25,644 (33.9) |
| High blood pressure/hypertension (yes), n (%) | 213,804 (70.5) | 65,673 (66.8) | 37,662 (74.0) | 52,535 (66.9) | 57,934 (76.7) |

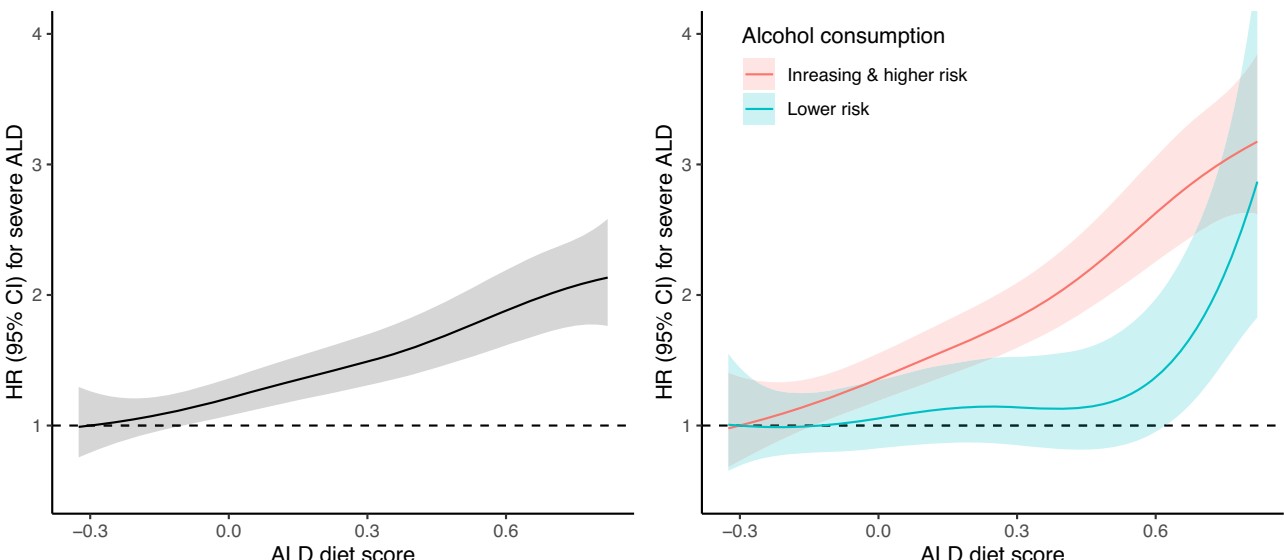

**Fig. 2 | Nonlinear association between diet score and severe ALD.** Data shown as hazard ratios (HRs) with their 95% confidence intervals (95% CI). Left panel: association in all participants with adjustment of total alcohol consumption; Right panel: association by alcohol consumption. All analyses were performed excluding people who developed the disease during the first two years of follow-up. Analyses were adjusted for age, sex, deprivation, ethnicity, the five components of the metabolic syndrome, frequency of alcohol consumption, smoking and physical activity. People with increasing and higher alcohol risk were grouped in this analysis because of insufficient numbers.

## Table 2 | Interaction between diet score and alcohol consumption categories on severe ALD risk

|  | HR (95% CI) | RERI (95% CI) | Multiplicative interaction (95% CI) |
|---|---|---|---|
| **Overall** |  | 2.44 (1.06–3.83) | 1.36 (0.88–2.11) |
| *Lower ALD diet score (<median)* |  |  |  |
| Lower alcohol risk | 1 (Reference) |  |  |
| Increasing alcohol risk | 2.28 (1.34–3.87) |  |  |
| Higher alcohol risk | 11.23 (6.24–20.21) |  |  |
| *Higher ALD diet score (≥median)* |  |  |  |
| Lower alcohol risk | 1.49 (1.02–2.18) |  |  |
| Increasing alcohol risk | 4.82 (2.99–7.75) |  |  |
| Higher alcohol risk | 14.20 (8.24–24.50) |  |  |
| **Women** |  | 3.22 (0.78–5.65) | 2.98 (1.34–6.61) |
| *Lower ALD diet score (<median)* |  |  |  |
| Lower alcohol risk (≤14 units) | 1 (Reference) |  |  |
| Increasing alcohol risk (15–34 units) | 1.89 (0.76–4.69) |  |  |
| Higher alcohol risk (≥35 units) | 4.90 (1.54–15.58) |  |  |
| *Higher ALD diet score (≥median)* |  |  |  |
| Lower alcohol risk (≤14 units) | 0.87 (0.48–1.59) |  |  |
| Increasing alcohol risk (15–34 units) | 4.13 (1.94–8.82) |  |  |
| Higher alcohol risk (≥35 units) | 10.67 (4.29–26.53) |  |  |
| **Man** |  | 2.23 (0.26–4.19) | 0.93 (0.53–1.66) |
| *Lower ALD diet score (<median)* |  |  |  |
| Lower alcohol risk (≤14 units) | 1 (Reference) |  |  |
| Increasing alcohol risk (15–49 units) | 2.77 (1.38–5.55) |  |  |
| Higher alcohol risk (≥50 units) | 14.04 (6.60–29.85) |  |  |
| *Higher ALD diet score (≥median)* |  |  |  |
| Lower alcohol risk (≤14 units) | 2.07 (1.23–3.50) |  |  |
| Increasing alcohol risk (15–49 units) | 5.87 (3.09–11.14) |  |  |
| Higher alcohol risk (≥50 units) | 16.29 (7.95–33.40) |  |  |

Data are presented as hazard ratio and their 95% confidence intervals. All analyses were performed excluding people who developed the disease during the first two years of follow-up. Analyses were adjusted for age, sex, deprivation, ethnicity, the five components of the metabolic syndrome, smoking and physical activity, and alcohol drinking frequency at baseline.
*RERI* relative risk due to interaction.

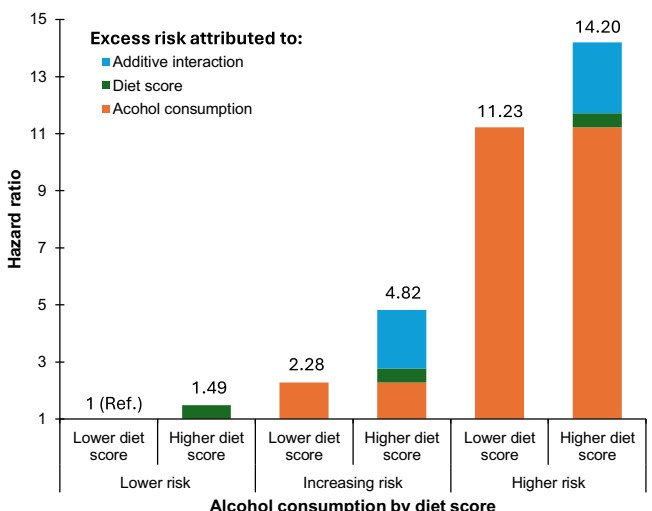

**Fig. 3 | Additive interaction of alcohol consumption and diet score with severe ALD.** Data shown as hazard ratios (HRs) attributed to different sources. All analyses were performed excluding people who developed the disease during the first two years of follow-up. Analyses were adjusted for age, sex, deprivation, ethnicity, the five components of the metabolic syndrome, frequency of alcohol consumption, smoking and physical activity.

**Table 3 | Population attributable fraction of ALD diet score and alcohol consumption**

| | Prevalence, % | HR (95% CI) | Population attributable fraction, % (95% CI) |
|---|---|---|---|
| **ALD diet score** | | | 28.8 (16.0–39.6) |
| <median | 49.7 | 1 (Reference) | |
| ≥median | 50.3 | 1.59 (1.29–1.97) | |
| **Alcohol consumption** | | | 67.1 (61.5–71.9) |
| Lower risk | 58.3 | 1 (Reference) | |
| Increasing risk | 34.5 | 3.27 (2.11–5.08) | |
| Higher risk | 7.2 | 11.34 (7.11–18.08) | |

Population attributable fractions assume the HR shown to be causal which could not be shown in this study. All analyses were performed excluding people who developed the disease during the first two years of follow-up. Analyses were mutually adjusted, and additionally for age, sex, deprivation, ethnicity, the five components of the metabolic syndrome, frequency of alcohol consumption, smoking and physical activity.

## Discussion

Using data from the UK Biobank study, we identified that a higher risk diet – measured through a data-driven score – amplified the association between alcohol consumption and severe ALD. In fact, people who had higher alcohol consumption and higher ALD diet score had a 14-fold higher risk of severe ALD compared with the 11-fold higher risk among people who had similar alcohol consumption but a lower diet score. This finding was corroborated by the additive interaction, with a relative excess risk due to an interaction of 2.4, indicating that having a higher ALD diet score and a higher alcohol consumption simultaneously confers to 2.4 times higher risk than the sum of excess risk of the two factors. Not surprisingly, the majority (67%) of severe ALD cases could be attributed to weekly alcohol consumption. However, our results also showed that a higher dietary score accounted for nearly 29% of cases when self-reported alcohol consumption was adjusted. While this study design cannot demonstrate causality, our findings illustrated the relevance of diet in severe ALD, either as a predictor or as an effect modifier of higher alcohol consumption.

Although the World Health Organisation (WHO) advises that neither women nor men should drink more than 20 g/day of alcohol, safe alcohol consumption levels are challenging to define[5]. A previous study carried out in 19 high-income countries identified that the threshold for alcohol to reduce the risk of all-cause mortality was about 100 g/week[23]. Recently, the WHO highlighted that no level of alcohol consumption does not have an adverse effect on health[24]. In our study, people who had higher ALD dietary risk score and ≥14 units of alcohol per week consumed, on average, 34.9 units/week, i.e., almost 1.5 times higher than the maximum safe alcohol consumption level advised by the WHO. Even if we highlighted small differences in consumption patterns between the high and low diet score groups, eating habits can be altered before, during, and following drinking episodes[25,26].

In terms of overall diet, people with alcohol use disorder may consume more energy from alcohol rather than food. In addition, as people with ALD may suffer from anorexia[27,28]. In the past, malnutrition used to be considered the leading cause of liver injury, before excessive alcohol consumption, and it is now recognised to be present in almost half of people with this condition[29]. Moreover, excess alcohol consumption can lead to reduced digestion and nutrition absorption through gut mechanical alterations[27–29]. In this context, malnutrition in people with ALD may result in reduced appetite, increased dysgeusia, and increased inflammations (characterised by increased blood concentrations of TNF-α, IL-1B, and IL8), leading to pancreatic insufficiency and gut epithelial damage[28].

The mechanisms through which diet could influence the progression to severe ALD incidence and pathology have been studied less. One possible mechanism by which diet may amplify alcohol-related liver injury is through excess fat intake. Despite the small number of participants ($n = 42$), a previous study found a correlation between high fat and oil consumption and elevated liver transaminase levels[30], which could partly explain the processed and red meat as selected by the LASSO algorithm in our study. A previous study highlighted that the hyperlipidaemia risk was higher among people with excessive meat and alcohol consumption. Therefore, the higher risk in our population may be partially explained by elevated transaminase levels associated with dyslipidaemia[31]. On the contrary, animal models have shown a potential protective role of saturated fats that were attributed to the modulation of the hepatic sirtuin−Sterol Regulatory Element Binding Protein-1c−histone H3 axis, which may reduce expression of genes encoding lipogenic enzymes and, therefore, the induction of adiponectin[32]. Conversely, studies in the same animal models have shown an adverse effect of unsaturated fats in promoting alcohol-induced liver damage, that was attributed, at least in part, to increased levels of pro-inflammatory oxidised linoleic acid metabolite[33]. High salt intake, another factor in the diet score, was found to be associated with higher liver fibrosis through excess reactive oxygen species production[34] and increased fluid accumulation[35], which may speed up the severe phase of the disease. On the contrary, while refined grain consumption increases the likelihood of liver disease, whole grains, commonly found in cereals, have been associated with lower abdominal fat levels and inflammation[36].

Our study shows that the combination of four specific dietary items exacerbates the risk of severe ALD due to alcohol consumption. We should, however, note that the diet score selected by LASSO was not meant to be causal but predictive of severe ALD. The diet score may indicate other causal factors (e.g., drinking patterns) which interact with alcohol consumption even though average drinking frequency was adjusted for. It could also be a proxy of the imprecision of self-reported alcohol consumption.

There has been extensive investigation of the role of volume and pattern of alcohol consumption in the development of ALD (especially cirrhosis) in previous prospective studies[8,9]. For instance, Askgaard

et al. showed that daily drinking was associated with a higher alcohol-related cirrhosis risk, especially in men. Compared to men and women who drank fewer than 14 units per week, current abstainers had 7.58 (95% CI: 3.39–16.9) and 3.21 (95% CI: 0.77–13.4) times higher risk[8], respectively. Another study conducted in the UK established that alcohol-related cirrhosis risk was higher in women who drank alcohol daily (RR: 1.61 [95% CI: 1.40–1.85]); however, the risk was almost 2.5 times higher in women who drank alcohol without food[9]. These findings indicate the relevance of food in the development of severe ALD since women who drank alcohol with meals had a lower risk of alcohol-related cirrhosis[9]. However, to our knowledge, the joint association of diet and alcohol in influencing ALD risk has not been investigated previously. Moreover, the Clinical Practice Guideline on managing alcohol-related disease has not included diet as part of the recommendations[37]. Therefore, our findings reinforce the need to study diet and other risk factors that may independently, and/or jointly, modify liver-related outcomes.

Our research question was investigated in a single, large, and well-characterised general population cohort of middle-aged and older adults. Analyses were adjusted for a comprehensive set of covariates. A major driver of potential information bias, knowledge of disease status, was obviated entirely by ascertaining outcomes from routine administrative databases. Finally, because of the lack of consensus on which dietary items are associated with ALD risk, we used a data-driven approach to identify the dietary items to construct the diet score. Nonetheless, this study also has limitations. First, although we included those confounding factors that were considered relevant and for which we had data, residual confounding due to unknown or unmeasured confounders is possible. The associations observed in this observational study cannot be assumed to infer causality. Specifically, this study could not suggest whether or not diet is a causal factor of ALD and, as such, the population attributable fractions should only be regarded as an indicative estimate. In addition, some of the adjusted variables, notably the components of metabolic syndrome, could be mediators, and the adjustment of these could underestimate the association of diet and alcohol with ALD. Second, alcohol consumption, the diet score, as well as some other covariates, were based on self-reported data, which may result in some inaccuracies. Diet is subject to recall and misclassification bias and may change over time. We attempted to minimise potential reverse causation by using a 2-year landmark analysis. Third, there were no detailed drinking pattern data (e.g., binge drinking) in the whole UK Biobank cohort, and future studies should investigate any interactions between alcohol drinking patterns and diet. However, we did adjust for the average frequency of alcohol drinking which could be a proxy of drinking pattern. Fourth, ascertainment of severe ALD was based on hospital admission and death records and, therefore, was restricted to more advanced or severe cases of the disease and should be interpreted accordingly. Fifth, the dietary items included in the ALD diet score were modestly correlated ($r < 0.3$) and, therefore, the coefficients estimated should not be regarded as a standard. This could affect the accuracy of the diet score and thus underestimate the association between diet score and severe ALD as a form of regression dilution bias[38]. Finally, UK Biobank does not represent the UK population regarding lifestyle and prevalent diseases. Therefore, whilst risk estimates might be generalisable[39], summary statistics such as prevalence and incidence might not be[40]. Importantly, dietary items selected from this study are generated using a data-driven method and should be externally validated in other datasets.

In summary, a higher ALD diet score was associated with severe ALD incidence after adjusting for total alcohol consumption and frequency of drinking. Moreover, there was an additive interaction between diet score and alcohol consumption, suggesting that a higher diet score could amplify the adverse effect of alcohol consumption on severe ALD. If these findings are corroborated in other datasets and proven causal, relevant public health and clinical guidelines should consider including diet advice to reduce the risk of severe ALD.

## Methods

UK Biobank was approved by the North West Multi-Centre Research Ethics Committee (Ref: 11/NW/0382). The study protocol is available online (http://www.ukbiobank.ac.uk/). This work was conducted under the UK Biobank application number 71392. UK Biobank is a prospective cohort study that enrolled over 500,000 participants aged 37–73 years from the general UK population, with a 5.5% response rate[41]. From 2006 to 2010, these participants visited one of 22 assessment centres in Scotland, England, and Wales. During their initial visit, participants completed a questionnaire on a touchscreen device, underwent physical measurements, and provided biological samples[42,43]. More information about the UK Biobank protocol can be found online (http://www.ukbiobank.ac.uk).

### Severe alcohol-related liver disease

Severe ALD was defined as hospitalisation or death and was ascertained from the linked hospital and death databases during the follow-up. While there is no direct observation of ALD severity, these cases are assumed to be severe because a milder form of ALD is managed in the primary care setting. The date and cause of death were obtained from death certificates held by the National Health Service (NHS) Information Centre (England and Wales) and the NHS Central Register Scotland. Dates and causes of hospital admissions were identified through record linkage to Health Episode Statistics (England and Wales) and the Scottish Morbidity Records 01. Details of the linkage procedure are available at http://content.digital.nhs.uk/services. Hospital admissions and mortality data were available until the end of October 2021. Therefore, follow-up was censored on these dates, or until the date of death or hospitalisation for ALD if these occurred earlier.

Using the International Classification of Diseases, 10th revision (ICD-10), and an Expert Panel Consensus Statement[44], severe ALD was defined as ICD-10 K70. For sensitivity analyses, alcoholic-related hepatitis (K70.1) and alcoholic-related cirrhosis (K70.3) were also examined.

### ALD diet score

The self-completed touchscreen questionnaire, which was completed at baseline, provided data on the frequency of dietary item consumption over the previous year. Of the 29 questions related to dietary intake, 9 items were excluded because they were not deemed to be related to ALD or were superseded by other variables. These included: (1) hot drink temperature, (2) water intake, (3) coffee type, (4) age when last ate meat, (5) never eat eggs, dairy, wheat, and sugar, (6) non-butter spread type details, (7) a pilot version of (5), (8) a pilot version of bread type, and (9) a pilot version of spread type. This resulted in the analysis of 20 dietary items: cooked vegetable, salad/raw vegetable, fresh fruit, dried fruit, oily fish, non-oily fish, processed meat, poultry, beef, lamb, pork, cheese, milk type used, spread type, bread type, cereal intake, cereal type, salt added to food, tea, and coffee. Detailed information about these variables and the data collection and reporting methods have been published elsewhere[45,46].

The 20 dietary items were coded as ordinal variables and included in the regularised Cox regression models using the LASSO (Least Absolute Shrinkage and Selection Operator) algorithm. LASSO is a form of regularisation that reduces the coefficient so that the prediction would be more accurate in external data[47]. The coefficients derived from this analysis are available in Supplementary Fig. 1. The interrelationship among these 20 dietary items, as well as their individual association with the outcome of interest, are shown in Supplementary Fig. 2 and Supplementary Table 2.

Subsequently, a cross-validation analysis was performed to compute the 10-fold cross-validation for Cox Models. This was done to

predict which dietary items should be included in the final diet score while minimising the risk of overfitting. This validation aimed to maximise Harrel´s C-index, where a higher C-index indicates superior risk discrimination performance. Supplementary Fig. 3 shows the cross-validated C-index for the optimal λ values (the strength of regularisation) of the food elements included. The left vertical line in the figure shows where the CV-error curve reached its minimum, while the right vertical line shows the most regularised model with CV-error within one standard deviation (SD) of the minimum. Four dietary items achieved the minimum error (Supplementary Table 3) and were used to create the score: processed meat, beef, cereal, and salt intake.

A continuous diet score was then created as the sum of each of the four dietary items multiplied by their corresponding coefficient values in the final model (Supplementary Table 2). The score roughly follows the normal distribution with a mean of 0.14 (0.34), where a higher score indicates higher severe ALD risk.

## Units of alcohol

Alcohol consumption was reported as the number of servings of types of alcoholic beverages consumed and the units of alcohol per serving of alcoholic beverages using the methodology reported by Jani et al. [48]. The weekly units of alcohol consumption were categorised into: lower risk (≤14 units), increasing risk (15-34 units for women and 15-49 units for men), and higher risk (≥35 units for women and ≥50 units for men), as per the UK official guideline [49].

Finally, using the median intake of the diet score (0.079) and the units of alcohol per week (lower, increasing, and higher risk), the following categories were created: (1) lower ALD dietary score and lower alcohol risk; (2) lower ALD dietary score and increasing alcohol risk; (3) lower ALD dietary score and higher alcohol risk; (4) higher ALD dietary score and lower alcohol risk; (5) higher ALD dietary score and increasing alcohol risk; (6) higher ALD dietary score and higher alcohol risk. These categories were used owing to the statistical methods to compute additive interactions.

## Covariates

Age at baseline was determined from the date of birth and baseline assessment. Sex was self-reported at baseline. Deprivation (area-based socioeconomic status) was derived from the postcode of residence using the Townsend score [50]. Ethnicity was self-reported and categorised as: white and others. Self-reported smoking status was categorised as never, former or current smoker. The components of the metabolic syndrome – including central obesity, hyperglycaemia/diabetes, high blood pressure/hypertension, low HDL cholesterol, and high triglycerides – were identified using baseline data. Central obesity was defined as a waist circumference higher than 88 cm in women and 102 cm in men. Hyperglycaemia/diabetes was defined as fasting glucose ≥5.6 mmol/l or self-report of a physician diagnosis of diabetes. High blood pressure/hypertension was defined as a systolic blood pressure ≥130 mm Hg and/or a diastolic blood pressure ≥85 mm Hg or self-report of a physician diagnosis of hypertension. High triglycerides were defined as ≥1.7 mmol/l and low HDL cholesterol as <1.3 mmol/l in women and <1.0 mmol/l in men [51–53]. Finally, the level of physical activity was self-reported using the International Physical Activity Questionnaire short form [54]. Additional information on the measurements is available on the UK Biobank website (http://www.ukbiobank.ac.uk).

## Statistical analyses

Descriptive baseline characteristics, based on the joint diet and alcohol categories, are presented as means with SD for quantitative variables and as frequencies and percentages for categorical variables. Cohen's d and Phi coefficient were used to quantify the difference in alcohol consumption and frequency [55].

Nonlinear associations between the diet score and severe ALD were investigated using penalised cubic splines fitted in Cox proportional hazard models. The penalised spline is a variation of the basis spline, which is less sensitive to knot numbers and placements than restricted cubic splines [56]. For these splines, values were truncated at less than 5% and greater than 95%. To investigate if the association varied with alcohol intake, the analyses were repeated and stratified by alcohol consumption (≤ and >14 units per week). Higher alcohol risk was not stratified because of insufficient numbers in that category.

The joint association of diet score (dichotomised using the median) and alcohol consumption (lower, increasing and higher risk) with ALD were investigated using Cox proportional hazard models. The results are presented as hazard ratios (HR) and their 95% confidence intervals (95% CIs). Both multiplicative and additive interactions were examined. The product term and the relative excess risk due to interaction (RERI) were employed to investigate multiplicative and additive interactions, respectively. It has been suggested that additive interactions might better capture biological interactions [57]. This analysis was performed for the overall population as well as by sex.

Individuals classified in the category with a lower ALD dietary risk score for ALD and consuming <14 units of alcohol were used as the referent group. The proportional hazard assumptions were verified using Schoenfeld residuals, with the duration of follow-up employed as the time-dependent variable. As sensitivity analyses, two stages of ALD progression were used as separate outcomes: alcoholic-related hepatitis and alcoholic-related cirrhosis. Moreover, alcohol consumption as a continuous variable and scaled per 7 units/weeks was used in calculating interactions as that might capture the influence of alcohol consumption on severe ALD risk better.

Based on the Expert Panel Consensus Statement [44] (Supplementary Table 1), participants with liver diseases or alcohol/drug use disorders at/or before baseline (n = 70,058), those with missing data for the created diet score (the four dietary elements included, n = 23,947) and those with missing data for one or more covariates (n = 105,011) were excluded. In addition, all analyses were conducted using 2-year landmark analyses, excluding all participants who experienced events within the first two years of follow-up (n = 80) (Fig. 1). This approach minimised the risk of reverse causality.

Analyses were adjusted for confounding factors based on previous literature, adjusted for sociodemographic factors (age, sex, deprivation and ethnicity), lifestyle factors (smoking and physical activity), health-related factors (components of the metabolic syndrome: central obesity, hyperglycaemia/diabetes, hypertension/high blood pressure, low HDL cholesterol and high triglycerides), as well as the frequency of alcohol drinking, which serve as a proxy for alcohol drinking pattern. Components of metabolic syndrome could be confounders or mediators but were adjusted to provide conservative estimates. The analysis of the association between the diet score and severe ALD was also adjusted for alcohol consumption (units/week).

Finally, the population attributable fraction (PAF) was calculated to estimate the proportion of ALD cases attributable to the diet score and alcohol consumption separately [58].

Stata 18 (StataCorp LLC) and R 4.3.1 (The R Foundation for Statistical Computing) were used to perform the analyses. A two-sided p-value below 0.05 was considered statistically significant.

## Reporting summary

Further information on research design is available in the Nature Portfolio Reporting Summary linked to this article.

## Data availability

Data cannot be shared directly due to the material transfer agreement from UK Biobank. All UK Biobank information is available online on the webpage https://www.ukbiobank.ac.uk/. Data access is available through applications. This research was conducted using the application number 71392.

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

## Acknowledgements

The study was supported by the British Heart Foundation (RE/18/6/34217). The authors are grateful to the UK Biobank participants. UK Biobank was established by the Wellcome Trust medical charity, the Medical Research Council, the Department of Health, the Scottish Government and the Northwest Regional Development Agency. It has also had funding from the Welsh Assembly Government and the British Heart Foundation. All authors had final responsibility for submission for publication.

## Author contributions

F.P-R, E.F. and F.K.H contributed to the conception and design of the study, advised on all statistical aspects, and interpreted the data. F.P.-R. performed the literature search. F.P.-R and F.K.H. performed the analyses and wrote the first draft. Z.Z., J.C.M., C.C.-M. D.R., N.S., and J.P.P. interpreted the data and critically revised the drafts. All authors approved the final draft for submission. F.P.-R., E.F and F.K.H contributed equally to this work and are joint senior authors. F.K.H. is the guarantor.

## Competing interests

N.S. declares consulting fees and/or speaker honoraria from Abbott Laboratories, Afimmune, Amgen, AstraZeneca, Boehringer Ingelheim, Eli Lilly, Hanmi Pharmaceuticals, Janssen, Merck Sharp & Dohme, Novartis, Novo Nordisk, Pfizer, and Sanofi; and grant support paid to his university from AstraZeneca, Boehringer Ingelheim, Novartis, and Roche Diagnostics. None of these disclosures are directly related to the study, nor its conception, analyses or interpretation. The other authors declare no conflict of interest.
