## [Peer Review File · Nature Communications]

Diet modifies the association between alcohol consumption and severe alcohol-related liver disease incidenceREVIEWER COMMENTS

Reviewer #1 (Remarks to the Author):

The study analysed the influence of diet on the incidence of ALD stratified with alcohol consumption of <14 units of alcohol and >14 units of alcohol. The broader findings are not surprising as it has been known that diet influences the incidence and progression of ALD.

However, the strength of the study is the use of data-driven approach to identify the food items to construct the diet score, and use it to analyse the incidence of ALD in low vs high alcohol consumption setting. The other noteworthy aspect is that the study used risk mitigation for reverse causation by utilising a 2-year landmark analysis.

Use of the well characterised and large UK Biobank cohort for analysis provides confidence in the study results. The manuscript is written with clarity and meets the ethical requirement.

The methodology is sound, meeting the expected standards in the field. The approach to test their hypothesis and interpretation of results are appropriate.

The findings are of significance in the field, academically and have clinical relevance.

Minor comments

Lines 142 & 144: UK Biobank has genetic data on sex and ethnicity which would have been more accurate than self report for both sex and ethnicity determination.

Can also add more recent reference showing coffee consumption protects ALD in UK Biobank cohort

Whitfield et al Am J Gastroenterol 2021

Minor typos in abstract: capitalise Using (line 38) and add i) (line 39)

Reviewer #2 (Remarks to the Author):

Related to the MS entitled "Diet modifies the association between alcohol intake and alcohol-related liver disease: A prospective study from UK Biobank, the article falls short of the quality standards required for publication in Nature Communications. The significance of diet in alcoholic liver disease (ALD) remains unclear, as the study solely relies on alcohol consumption and baseline diet, which may change throughout the follow-up period.

Moreover, by setting the alcohol consumption threshold at less than 14 units per week, it's difficult to ascertain if the patient truly has ALD. The analysis doesn't clearly delineate between the risk of developing ALD and the risk of morbidity and mortality associated with it, indicating a lack of conceptual clarity.

Points:

1. "Alcohol consumption is the leading risk factor of alcohol-related liver disease (ALD) but it is elusive why some people are more susceptible"- This sentence is incorrect. Alcohol is the necessary factor for the existence of ALD, whether it's past or current consumption. Subsequently, the risk of progression depends on genetic susceptibility and other factors (metabolic, environmental, etc.).

2. "A poor diet was associated with ALD incidence independent of alcohol consumption. People who have a poor diet are susceptible to ALD risk due to alcohol consumption." This is incorrect. The assertion of independence from alcohol consumption in a liver disease primarily caused by alcohol is implausible and contradicts the findings presented in the study.

3. "However, few studies have delved into the influence of diet (either in terms of individual or combined foods and nutrients) on ALD risk". This is incorrect. It's not just about risk, it's about progression. Alcohol consumption creates the risk, while other factors may influence the progression.

4. "Incident ALD was defined as hospitalisation or death and was ascertained from the linked hospital and death databases during the follow-up" - This is incorrect. Firstly, they need to specify how they have defined ALD, followed by clarification on the "incident" of ALD. Regarding the incident of ALD, it is essential to clarify whether they have solely considered causes of death and hospitalization attributed to ALD (as it is unclear in the writing and could include cases unrelated to alcohol consumption, such as accidents). If the

incidents are due to ALD-related causes, they should specify which causes, such as decompensation of cirrhosis, acute alcoholic hepatitis, and others.

5. The p value is missing in the table I

6. Fig 3 The title for the y-axis: "HR for severe NAFLD?"

7. This sentence is not correct: There has been extensive investigation of the role of alcohol consumption in development ALD (especially cirrhosis) in previous prospective studies 8,9. The focus of these studies was on the pattern of alcohol consumption rather than the consumption itself. The authors maintain a conceptual error that persists throughout the entire article.

Reviewer #3 (Remarks to the Author):

This manuscript describes the joint associations of alcohol intake and dietary factors with risk of alcohol-related liver disease (ALD), based on data in just over 300,000 participants (and 539 incident ALD cases) from the UK Biobank. This is an important topic given the increasing global disease burden due to ALD. The authors have generated a dietary score associated with ALD, using a LASSO algorithm to identify self-reported dietary questionnaire items related to higher ALD risk. They have reported that there is a potential additive interaction between this dietary score, and higher alcohol intake, to increase ALD risk. The manuscript is generally clearly written, however, there is insufficient information provided to fully evaluate the methods used and reproduce the findings. Although the authors have identified a research gap, the findings presented are minimal, and the scope of the study would need to be substantially expanded in order to make a contribution to advancing the field.

1. The association of the dietary score with ALD seems to be circular. These variables were selected and weighted based on their association with ALD (methods), and hence this score associates with ALD (Figure 1). Can the findings be validated in an independent dataset?

2. What is the relationship between the dietary score and alcohol intake? Do patterns of alcohol intake (e.g. weekly units, binge, types, with food, daily) differ between the high and low diet score groups? This could be a factor contributing to the excess risk (RERI) observed

in the high diet score/high alcohol intake group. In previous studies, particular drinking patterns have been shown to be important for liver disease risk, but the present study only seems to consider drinking below or above 14 units/week.

3. In the analysis of diet score and ALD, how was alcohol intake adjusted for (p10 line 197)? As confounding is particularly relevant for studies of diet, details on the other covariate adjustments should also be given in more detail.

4. The manuscript describes the dietary score in terms of “unhealthy diet” or “poor diet”. However, this score is based on four dietary items associated with ALD risk, rather than a “poor diet” per se. There is little mention of the actual dietary items involved in the score (beef, processed meat, cereal intake-inverse, salt) which are only shown in Supplementary table 2. These 4 items are not discussed in terms of potential mechanisms of action, or public health relevance.

5. The methods state that 9 diet items were excluded because they did not align with the study objectives (p6 line 106-108). What criteria were used to exclude these items? What are the excluded items?

6. The details of the ordinal variables used for the 20 included diet items needs to be explained – how many levels and what range of food intake frequency do they cover, and how does this relate to the score? How are the dietary items correlated with each other, and with alcohol and covariates? Which of the individual diet items are associated with ALD using standard adjusted regression models?

7. Supplementary figure 1 needs detailed explanation in the figure legend. What variables do each of the coloured lines represent? Supplementary figure 2 also needs an explanatory figure legend.

8. The number of ALD cases included in the study was 539, after median follow-up of 10.7 years. Could this be higher if further follow-up is included, according to what is available in UKB? What are the case numbers in men and women? Given the difference in drinking

patterns between men and women in the study, can subgroup analysis be performed? It would be helpful to include case numbers in all the relevant tables (e.g. by the four exposure groups in Table 2) and Figures.

9. Why is the ALD PAF for alcohol only 50.9 % (Table 3)? How was this calculated? It is surprising that the PAF from the diet score is close to that of alcohol. The methods (p10 line 198-200) state that PAFs were derived using adjusted HRs from nonlinear associations, but these analyses are not reported for alcohol intake.

Reviewer #1 (Remarks to the Author):

1. The study analysed the influence of diet on the incidence of ALD stratified with alcohol consumption of <14 units of alcohol and >14 units of alcohol. The broader findings are not surprising as it has been known that diet influences the incidence and progression of ALD. However, the strength of the study is the use of data-driven approach to identify the food items to construct the diet score, and use it to analyse the incidence of ALD in low vs high alcohol consumption setting. The other noteworthy aspect is that the study used risk mitigation for reverse causation by utilising a 2-year landmark analysis. Use of the well characterised and large UK Biobank cohort for analysis provides confidence in the study results. The manuscript is written with clarity and meets the ethical requirement. The methodology is sound, meeting the expected standards in the field. The approach to test their hypothesis and interpretation of results are appropriate. The findings are of significance in the field, academically and have clinical relevance.

Response: We thank the reviewers and editors for their comments. We hope our changes have improved the document.

Minor comments

2. Lines 142 & 144: UK Biobank has genetic data on sex and ethnicity which would have been more accurate than self report for both sex and ethnicity determination.

Response: Thank you for the suggestion. However, we do not think this is required for this study. Firstly, biological sex is well characterised in this cohort and transgender is very uncommon for this age group. Secondly, genetically predicted ethnicity has been shown to be inaccurate*. What the UK Biobank captured as ethnicity is essentially a self-identified cultural concept rather than based on biology. Regardless, we acknowledge that some measures will suffer from reporting bias which we have acknowledged as a limitation.

* *Reference:* Manica A, Prugnolle F, Balloux F. Geography is a better determinant of human genetic differentiation than ethnicity. *Human genetics*. 2005 Dec;118:366-71.

3. Can also add more recent reference showing coffee consumption protects ALD in UK Biobank cohort
Whitfield et al *Am J Gastroenterol* 2021

Response: Thank you for this suggestion. This study was included in the Introduction.

4. Minor typos in abstract: capitalise Using (line 38) and add i) (line 39)

Response: Thank you for bringing this to our attention. We have removed this part from abstract due to the word limit.

Reviewer #2 (Remarks to the Author):

1. Related to the MS entitled “Diet modifies the association between alcohol intake and alcohol-related liver disease: A prospective study from UK Biobank, the article falls short of the quality standards required for publication in Nature Communications. The significance of diet in alcoholic liver disease (ALD) remains unclear, as the study solely relies on alcohol consumption and baseline diet, which may change throughout the follow-up period. Moreover, by setting the alcohol consumption threshold at less than 14 units per week, it's difficult to ascertain if the patient truly has ALD. The analysis doesn't clearly delineate between the risk of developing ALD and the risk of morbidity and mortality associated with it, indicating a lack of conceptual clarity.

Response: We appreciate the reviewer’s careful observation and comments throughout. These are very important in ensuring our manuscript to be as accurate as possible. We agree that our study has limitations in the areas that were stated, primarily on the potential omission of milder ALD that does not require hospitalisation. In this regard, we have reworded the title, and throughout the manuscript, to indicate these outcomes are ‘Severe ALD’. However, we would like to highlight that the definition of Severe ALD using ICD-10 codes is based on a previously published consensus (which was included in the method section)* and should be as robust as routine healthcare data could provide.

*Reference: Hagström H, Adams LA, Allen AM, Byrne CD, Chang Y, Grønbaek H, et al. Administrative Coding in Electronic Health Care Record-Based Research of NAFLD: An Expert Panel Consensus Statement. *Hepatology*. 2021;74(1):474-82.

2. “Alcohol consumption is the leading risk factor of alcohol-related liver disease (ALD) but it is elusive why some people are more susceptible”- This sentence is incorrect. Alcohol is the necessary factor for the existence of ALD, whether it's past or current consumption. Subsequently, the risk of progression depends on genetic susceptibility and other factors (metabolic, environmental, etc.).

Response: Thank you for your comment which we agree. This line has been rewritten to more accurately reflect this as well as this study is looking at Severe ALD.

3. “A poor diet was associated with ALD incidence independent of alcohol consumption. People who have a poor diet are susceptible to ALD risk due to alcohol consumption.· this is incorrect. The assertion of independence from alcohol consumption in a liver disease primarily caused by alcohol is implausible and contradicts the findings presented in the study.

Response: Thank you for this observation. We have now revised the line to more accurately reflect the study examined Severe ALD – a more severe form of ALD.

4. . “However, few studies have delved into the influence of diet (either in terms of individual or combined foods and nutrients) on ALD risk”. This is incorrect. It's

not just about risk, it's about progression. Alcohol consumption creates the risk, while other factors may influence the progression.

Response: We agree that the role of diet is on ALD progression rather than the onset, which should be due to alcohol consumption. We have revised the sentence for this fact.

5. "Incident ALD was defined as hospitalisation or death and was ascertained from the linked hospital and death databases during the follow-up"- This is incorrect. Firstly, they need to specify how they have defined ALD, followed by clarification on the "incident" of ALD. Regarding the incident of ALD, it is essential to clarify whether they have solely considered causes of death and hospitalization attributed to ALD (as it is unclear in the writing and could include cases unrelated to alcohol consumption, such as accidents). If the incidents are due to ALD-related causes, they should specify which causes, such as decompensation of cirrhosis, acute alcoholic hepatitis, and others.

Response: Sorry for the confusion. Our study focuses on severe ALD, which is a progression from the milder form of ALD. This has been clarified in the manuscript throughout.

5. The p value is missing in the table I

Response: Thank you for your comment. However, we do not include p-values in Table 1 because those are not the aim of the study. Instead, Table 1 is for the providing the average characteristics of people in each category. This is in line with the STROBE guideline that advises against the use of inferential statistics in descriptive tables*.

**Reference:* Vandembroucke JP, Elm EV, Altman DG, Gøtzsche PC, Mulrow CD, Pocock SJ, Poole C, Schlesselman JJ, Egger M, Strobe Initiative. Strengthening the Reporting of Observational Studies in Epidemiology (STROBE): explanation and elaboration. *Annals of internal medicine.* 2007 Oct 16;147(8):W-163.

6. Fig 3 The title for the y-axis: "HR for severe NAFLD?"

Response: Apologies for the confusion. We have now corrected the typo in the Figure. Moreover, all figures were modified to included in the title "Severe ALD".

7. This sentence is not correct: There has been extensive investigation of the role of alcohol consumption in development ALD (especially cirrhosis) in previous prospective studies 8,9. The focus of these studies was on the pattern of alcohol consumption rather than the consumption itself. The authors maintain a conceptual error that persists throughout the entire article.

Response: Thank you for pointing out the imprecision in the description. We have revised the statement to reflect that those studies are on the pattern of consumption rather than just the consumption itself.

Reviewer #3 (Remarks to the Author):

This manuscript describes the joint associations of alcohol intake and dietary factors with risk of alcohol-related liver disease (ALD), based on data in just over 300,000 participants (and 539 incident ALD cases) from the UK Biobank. This is an important topic given the increasing global disease burden due to ALD. The authors have generated a dietary score associated with ALD, using a LASSO algorithm to identify self-reported dietary questionnaire items related to higher ALD risk. They have reported that there is a potential additive interaction between this dietary score, and higher alcohol intake, to increase ALD risk. The manuscript is generally clearly written, however, there is insufficient information provided to fully evaluate the methods used and reproduce the findings. Although the authors have identified a research gap, the findings presented are minimal, and the scope of the study would need to be substantially expanded in order to make a contribution to advancing the field.

Response: We thank the reviewers and editors for their comments. We hope our changes have improved the document.

1. The association of the dietary score with ALD seems to be circular. These variables were selected and weighted based on their association with ALD (methods), and hence this score associates with ALD (Figure 1). Can the findings be validated in an independent dataset?

Response: Thank you for your suggestion. We appreciate the findings of this study requires external validation. However, we do not have access to another dataset that have all the required variables and therefore would not be able to conduct such validation. We have acknowledged this as a limitation.

2. What is the relationship between the dietary score and alcohol intake? Do patterns of alcohol intake (e.g. weekly units, binge, types, with food, daily) differ between the high and low diet score groups? This could be a factor contributing to the excess risk (RERI) observed in the high diet score/high alcohol intake group. In previous studies, particular drinking patterns have been shown to be important for liver disease risk, but the present study only seems to consider drinking below or above 14 units/week.

Response: Thank you for your question. Participants' characteristics including alcohol intake frequency and total units/week are now shown in Table 1. We also quantified the difference using effect size measures. There were small differences in terms of consumption and consumption pattern between the high and low diet score groups (Cohen's d 0.09 for total consumption; Phi coefficient 0.12 for frequency of drinking). We have now made these comparisons explicit in the Results and discussed them as potential mechanism in the paper. Unfortunately, we only have data on the weekly intake of alcohol rather than the intake per once occasion (which is the typical definition of binge drinking). However, we would like to argue that people who are in the high consumption categories are also more like to have binge drinking episodes, and a previous study that showed total consumption has a much stronger association with

ALD than drinking pattern (Simpson et al, 2019)*. The needs for research on diet and drinking pattern is now discussed.

**Reference: Simpson RF, Hermon C, Liu B, Green J, Reeves GK, Beral V, Floud S. Alcohol drinking patterns and liver cirrhosis risk: analysis of the prospective UK Million Women Study. The Lancet Public Health. 2019 Jan 1;4(1):e41-8.*

3. In the analysis of diet score and ALD, how was alcohol intake adjusted for (p10 line 197)? As confounding is particularly relevant for studies of diet, details on the other covariate adjustments should also be given in more detail.

Response: Thank you for raising this point. We have now provided the adjustment models in the descriptions of results.

4. The manuscript describes the dietary score in terms of “unhealthy diet” or “poor diet”. However, this score is based on four dietary items associated with ALD risk, rather than a “poor diet” per se. There is little mention of the actual dietary items involved in the score (beef, processed meat, cereal intake-inverse, salt) which are only shown in Supplementary table 2. These 4 items are not discussed in terms of potential mechanisms of action, or public health relevance.

Response: Thank you for these suggestions. We did not wish to implicate causality in the discussions as our study could not establish so. LASSO selects variables based on prediction. However, we understand the reviewer’s concern and have speculated some potential mechanisms in the Discussions.

5. The methods state that 9 diet items were excluded because they did not align with the study objectives (p6 line 106-108). What criteria were used to exclude these items? What are the excluded items?

Response: We have clarified the excluded items because they were not deemed to be relevant to ALD in the manuscript. These are now clearly stated in the paper.

6. The details of the ordinal variables used for the 20 included diet items needs to be explained – how many levels and what range of food intake frequency do they cover, and how does this relate to the score? How are the dietary items correlated with each other, and with alcohol and covariates? Which of the individual diet items are associated with ALD using standard adjusted regression models?

Response: The details of the dietary items as well as their individual adjusted association with severe ALD are now expanded in the new Supplementary 2. There are coded as their responded categories and assumed to be linearly associated with risk in the LASSO model. The interrelationship of dietary items is shown in the new Supplementary Figure 2.

7. Supplementary figure 1 needs detailed explanation in the figure legend. What variables do each of the coloured lines represent? Supplementary figure 2 also needs an explanatory figure legend.

Response: We have added an explanatory legend to Supplementary Figures 1 and 2, including the legend for the coloured lines in Supplementary Figure 1. However, we found it difficult, and less relevant, to label other 14 variables in the Figure.

8. The number of ALD cases included in the study was 539, after median follow-up of 10.7 years. Could this be higher if further follow-up is included, according to what is available in UKB? What are the case numbers in men and women? Given the difference in drinking patterns between men and women in the study, can subgroup analysis be performed? It would be helpful to include case numbers in all the relevant tables (e.g. by the four exposure groups in Table 2) and Figures.

Response: Thank you for your observation. The data that we used are already up to date. The follow-up years had 2 years subtracted because of the 2-year landmark analysis. We do not envision that number of cases will increase dramatically if we wait for the next data release, which is due in the later half of this year. Regarding sex differences, we have added a subgroup analysis by sex on the interaction (Table 2).

9. Why is the ALD PAF for alcohol only 50.9 % (Table 3)? How was this calculated? It is surprising that the PAF from the diet score is close to that of alcohol. The methods (p10 line 198-200) state that PAFs were derived using adjusted HRs from nonlinear associations, but these analyses are not reported for alcohol intake.

Response: The outcome that we included is severe ALD, which is different from milder form of ALD that does not require hospitalisation. Conceptually all ALD should be attributed to alcohol (as reviewer 1 pointed out) however the progression to severe ALD is not necessarily. Our findings showed that the threshold of 14 units/week was only responsible for about half of the severe ALD cases, and that diet could capture some of the remaining progression through interaction with alcohol. The estimation for alcohol could be underestimate due to the use of binary variable. This is explicitly stated in the paper. The calculation of PAF is based on well-established formula that is used in the Global Burden of Diseases studies*.

*Reference: Mansournia MA, Altman DG. Population attributable fraction. *BMJ*. 2018 Feb 22;360.

REVIEWERS' COMMENTS:

Reviewer #1 (Remarks to the Author):

The authors have addressed the concerns of reviewers satisfactorily in their rebuttal and revised manuscript.

Reviewer #2 (Remarks to the Author):

the authors fulfil the requirements

Reviewer #3 (Remarks to the Author):

Thank you to the authors for revising their manuscript based on reviewers' comments. However, I still consider the scope and depth of the analyses and findings would need to be substantially expanded in order to address the research question and advance knowledge in the field.

The binary classification of alcohol intake into \leq 14 units per week will not capture the different ALD risks associated with alcohol intake at different levels or patterns of consumption, and also the different relationships between alcohol intake/patterns and the diet score in ALD risk. The UK Biobank data include more details on alcohol consumption which could be used by this study.

The diet score weighted by regression coefficients (Supp Table 2) includes correlated variables (Supp Fig 2): salt with beef and processed meat, beef with processed meat, cereal inverse with salt and processed meat. This could cause bias in the associations of the diet score with ALD.

The RRs used to calculate the alcohol and diet PAFs are not provided. It is still unclear why the PAF for the diet score is so high (40%) compared with the PAF for alcohol (51%) which the authors acknowledge may be underestimated due to the binary classification.

The manuscript refers to a “unhealthy/poor diet” but this was a diet score selected for association with ALD within the present study rather than being “unhealthy/poor” based on objective criteria.

The authors acknowledge that the diet score is not assessed for causality, and may in fact capture some elements related to alcohol drinking patterns, which could explain the study findings.

It is stated in the introduction that fatty liver is milder and cirrhosis is the advanced form of ALD. For this study the outcome definition is ICD-10 K70 (Methods) which includes fatty liver and other forms of ALD. Would the term hospitalised ALD be more appropriate than severe ALD?

In several places, the manuscript refers to progression to severe ALD. However, the analyses are based on incidence of ALD, rather than progression.

The covariate adjustments are not clear. Is the association of the diet score in Figure 2 adjusted for alcohol intake (not stated in results text, but it is in the figure legend). In Table 2 (footnote), it is unclear how alcohol intake at baseline was adjusted for as the categories are based on \leq 14 units/week.

The first sentence in the abstract refers to heavy drinkers, but the manuscript uses a cut-off of 14 units per week, which would not usually be described as heavy drinking.

Reviewer #3 (Remarks to the Author):

Thank you to the authors for revising their manuscript based on reviewers' comments. However, I still consider the scope and depth of the analyses and findings would need to be substantially expanded in order to address the research question and advance knowledge in the field.

Response: Thank you for reviewing our manuscript again. We have conducted several additional analyses as suggested.

The binary classification of alcohol intake into \leq 14 units per week will not capture the different ALD risks associated with alcohol intake at different levels or patterns of consumption, and also the different relationships between alcohol intake/patterns and the diet score in ALD risk. The UK Biobank data include more details on alcohol consumption which could be used by this study.

Response: Thank you for this suggestion. We have now conducted two additional sets of analyses, one with a 3-level categorical variable based on the UK NICE guideline to categorise alcohol risk (lower risk [0-14 units/week], increasing risk [15-34 for female; 15-49 for male], higher risk [all remaining]) – this replaces the original two-category analysis. The second additional analysis uses alcohol intake (units/week) as a continuous variable. The first set of analyses could provide an approximation for nonlinear interactions if any. The second set of analyses should provide a more accurate portrait of ALD hospitalisation risk due to very high alcohol intake. We have now also included the average frequency of alcohol drinking as a covariate in all analyses, which could serve as a proxy of binge drinking. This should be able to help triangulate the potential effect of binge drinking, even though the baseline questionnaire in the UK Biobank doesn't have data on each / max units in an occasion to derive an accurate measure of binge drinking. All results appear consistent with the original analysis, with various minor variations in the estimated RERI. We hope these could provide confidence to the robustness of the study's findings.

The diet score weighted by regression coefficients (Supp Table 2) includes correlated variables (Supp Fig 2): salt with beef and processed meat, beef with processed meat, cereal inverse with salt and processed meat. This could cause bias in the associations of the diet score with ALD.

Response: We understand the reviewer's concern over multicollinearity. However, we would like to suggest that these will not cause any severe bias in the results. Firstly, all correlations that the reviewer pointed out are of small effect size ($r < 0.3$). This means that if multicollinearity exists, the extent should be of small influence. Secondly, LASSO is one of the better approaches to handle multicollinearity (Bayo et al 2021) compared to ordinary regression models, resulting in smaller root mean squared errors. Thirdly, the suggested bias, if exists, will be manifested in the estimation of the coefficients of the diet score, because multicollinearity affects the estimated coefficients when correlated predictors are in the same model (i.e. the LASSO model). This means that these LASSO coefficients would be less accurate, resulting in a less strong association

between the diet score and ALD as a form of regression dilution bias. We have now put into the discussions.

Bayo AK, Rafiu AB, Funmilayo AT, Oluyemi OI. Investigating the impact of multicollinearity on linear regression estimates. *Malaysian Journal of Computing (MJoC)*. 2021;6(1):698-714.

The RRs used to calculate the alcohol and diet PAFs are not provided. It is still unclear why the PAF for the diet score is so high (40%) compared with the PAF for alcohol (51%) which the authors acknowledge may be underestimated due to the binary classification.

Response: We have now re-estimated the PAFs using the 3-level alcohol variable to provide a more accurate estimation of that of alcohol. Indeed, the PAF for alcohol is stronger (67%), while that for diet is weaker (28%), which agrees with the reviewer's suggestion. These are now reported in the paper. Discussions were also revised reflecting these.

The manuscript refers to a "unhealthy/poor diet" but this was a diet score selected for association with ALD within the present study rather than being "unhealthy/poor" based on objective criteria.

Response: Apologies for the incorrect use of description. We have now reworded all these as 'diet with higher/lower ALD risk'.

The authors acknowledge that the diet score is not assessed for causality, and may in fact capture some elements related to alcohol drinking patterns, which could explain the study findings.

Response: With the addition of a more refined categorical variable and a continuous variable, and the inclusion of the alcohol consumption frequency, the additive interaction presented should be robust.

It is stated in the introduction that fatty liver is milder and cirrhosis is the advanced form of ALD. For this study the outcome definition is ICD-10 K70 (Methods) which includes fatty liver and other forms of ALD. Would the term hospitalised ALD be more appropriate than severe ALD?

Response: We appreciate the thoughtful consideration of the most appropriate wording, and have considered rewording the outcome to be hospitalised ALD. However, we think this term would not be accurate because the outcome also included death with a cause of death including ALD – these included people who have never been hospitalised for it. We would like to highlight that milder form of ALD would not have been hospitalised as they are managed in primary care setting. Such interpretation has been used in our previous studies before in hepatology specific journals. We have made these clearer in the Methods.

Petermann-Rocha F, Gray SR, Forrest E, Welsh P, Sattar N, Celis-Morales C, Ho FK, Pell JP. Associations of muscle mass and grip strength with severe NAFLD: A prospective study of 333,295 UK Biobank participants. *Journal of Hepatology*. 2022 May 1;76(5):1021-9.

In several places, the manuscript refers to progression to severe ALD. However, the analyses are based on incidence of ALD, rather than progression.

Response: The outcome is the incidence of severe ALD. The use of progression to severe ALD corresponds to Reviewer 1's suggestion that the outcome we capture here is not the actual incidence of all ALD, but a severe form of it. However, to clarify the description, we have reworded 'progression to severe ALD' to 'onset of severe ALD'.

The covariate adjustments are not clear. Is the association of the diet score in Figure 2 adjusted for alcohol intake (not stated in results text, but it is in the figure legend). In Table 2 (footnote), it is unclear how alcohol intake at baseline was adjusted for as the categories are based on </> 14 units/week.

Response: Thank you for pointing out the inconsistency. We have now stated explicitly in the Results text that total alcohol consumption was adjusted for in Figure 2.

The first sentence in the abstract refers to heavy drinkers, but the manuscript uses a cut-off of 14 units per week, which would not usually be described as heavy drinking.

Response: Thank you for pointing out this inaccuracy. Since we have now reported findings with higher alcohol risk as per the UK NICE guideline, we feel that the description of 'heavy drinker' reflects the analysis.

REVIEWER COMMENTS

Reviewer #3 (Remarks to the Author):

Thank you to the authors for revising and improving their manuscript further based on reviewers' comments, and including some additional analyses and clarifications in the text.

I have the following comments/suggestions:

1. The analysis of alcohol as a continuous variable (Supplementary table 4) is unclear, as alcohol appears to have been grouped into 7 units/week lower vs 7 units higher groups, rather than analysed as a continuous variable with a resulting estimate per unit/week (or scaled otherwise). Please clarify in the Methods how this analysis was done.
2. The HR and PAF for the ALD diet score (Table 3) included adjustment for frequency of alcohol consumption – what is the effect of also adjusting for alcohol intake amount, which could remove possible further confounding between the diet score and alcohol intake.
3. The main analyses adjust for sociodemographic factors and five metabolic syndrome components – can a sensitivity analysis be conducted without the metabolic syndrome factors, as these may be potential mediators of the associations between diet/alcohol and ALD.
4. Figure 1: 105,011 participants are excluded due to missing data for covariates – a high proportion – the numbers with missing data for each covariate can be listed in this figure to provide more information.

REVIEWER COMMENTS

Reviewer #3 (Remarks to the Author):

Thank you to the authors for revising and improving their manuscript further based on reviewers' comments, and including some additional analyses and clarifications in the text.

I have the following comments/suggestions:

1. The analysis of alcohol as a continuous variable (Supplementary table 4) is unclear, as alcohol appears to have been grouped into 7 units/week lower vs 7 units higher groups, rather than analysed as a continuous variable with a resulting estimate per unit/week (or scaled otherwise). Please clarify in the Methods how this analysis was done.

Response: Thank you for your observation. The calculation of additive interaction from Cox models requires two examine distinct point on the continuous variables and therefore the presentation might be confusing. Information regarding this analysis was included at the bottom of the statistical analysis section as follows:

“As sensitivity analyses, two stages of ALD progression were used as separate outcomes: alcoholic-related hepatitis and alcoholic-related cirrhosis. Moreover, alcohol consumption as a continuous variable and scaled per 7 units/week was used in calculating interactions as that might capture the influence of alcohol consumption on severe ALD risk better.”

In addition, we have made a footnote in the Supplementary Table 4.

“Calculation of RERI based on Cox propositional model requires a comparison at two distinct points as the model was conducted on the multiplicative scale. In this analysis the two distinct points chosen were by 7-unit difference in alcohol intake.”

2. The HR and PAF for the ALD diet score (Table 3) included adjustment for frequency of alcohol consumption – what is the effect of also adjusting for alcohol intake amount, which could remove possible further confounding between the diet score and alcohol intake.

Response: Apologies for the confusion. The analyses in Table 3 are already mutually adjusted. That is, the results for ALD diet score was adjusted for consumption as well as other covariates listed. These are now made explicit in the footnote.

3. The main analyses adjust for sociodemographic factors and five metabolic syndrome components – can a sensitivity analysis be conducted without the metabolic syndrome factors, as these may be potential mediators of the associations between diet/alcohol and ALD.

Response: We recognise that these variables may act as potential mediators, as well as confounders (e.g. metabolic syndrome could initiate a clinician to discuss alcohol drinking with the person). We decided to keep them adjusted to provide a more conservative estimates. Nonetheless, we have discussed the potential influence of this (i.e. overadjustment / underestimation of the associations of alcohol and diet) in the Methods and Limitations.

4. Figure 1: 105,011 participants are excluded due to missing data for covariates – a high

proportion—the numbers with missing data for each covariate can be listed in this figure to provide more information.

Response: Thank you for this suggestion. The figure was updated with this new information.

REVIEWERS' COMMENTS

Reviewer #3 (Remarks to the Author):

Thank you for the revised manuscript addressing reviewer comments.

For Table 3, to clarify, the diet score PAF is now mutually adjusted for both alcohol consumption frequency and alcohol intake amount?

Please check typos (Supp. Table 4 footnote Cox propositional/proportional).

I have no further comments.

REVIEWER COMMENTS

1. Thank you for the revised manuscript addressing reviewer comments.
For Table 3, to clarify, the diet score PAF is now mutually adjusted for both alcohol consumption frequency and alcohol intake amount?

Response: Thank you for your question. Yes indeed, both alcohol consumption and frequency were adjusted for the calculation of PAF for diet.

2. Please check typos (Supp. Table 4 footnote Cox propositional/proportional).

Response: Thank you for bringing this to our attention. This was corrected.